# Pre-Operative MDCT Staging Predicts Mesopancreatic Fat Infiltration—A Novel Marker for Neoadjuvant Treatment?

**DOI:** 10.3390/cancers13174361

**Published:** 2021-08-28

**Authors:** Sami-Alexander Safi, Lena Haeberle, Sophie Heuveldop, Patric Kroepil, Stephen Fung, Alexander Rehders, Verena Keitel, Tom Luedde, Guenter Fuerst, Irene Esposito, Farid Ziayee, Gerald Antoch, Wolfram Trudo Knoefel, Georg Fluegen

**Affiliations:** 1Department of General, Visceral, Thoracic and Pediatric Surgery (A), Medical Faculty, Heinrich-Heine-University and University Hospital Duesseldorf, Moorenstraße 5, 40225 Düsseldorf, Germany; sami-alexander.safi@med.uni-duesseldorf.de (S.-A.S.); Stephen.Fung@med.uni-duesseldorf.de (S.F.); Rehders@med.uni-duesseldorf.de (A.R.); knoefel@hhu.de (W.T.K.); Georg.Fluegen@med.uni-duesseldorf.de (G.F.); 2Institute of Pathology, Medical Faculty, Heinrich-Heine-University and University Hospital Duesseldorf, Moorenstraße 5, 40225 Düsseldorf, Germany; LenaJulia.Haeberle@med.uni-duesseldorf.de (L.H.); Irene.Esposito@med.uni-duesseldorf.de (I.E.); 3Department of Diagnostic and Interventional Radiology, Medical Faculty, Heinrich-Heine-University and University Hospital Duesseldorf, Moorenstraße 5, 40225 Düsseldorf, Germany; sophie.heuveldop@gmx.de (S.H.); Patric.kroepil@hhu.de (P.K.); Fuerst@med.uni-duesseldorf.de (G.F.); Antoch@med.uni-duesseldorf.de (G.A.); 4Department of Gastroenterology, Hepatology and Infectious Diseases, Medical Faculty, Heinrich-Heine-University and University Hospital Duesseldorf, Moorenstraße 5, 40225 Düsseldorf, Germany; Verena.Keitel@med.uni-duesseldorf.de (V.K.); Tom.Luedde@med.uni-duesseldorf.de (T.L.)

**Keywords:** PDAC, mesopancreas, fat stranding, radiographic imaging, MDCT

## Abstract

**Simple Summary:**

After the implementation of an internationally recognized histopathological protocol, the rate of complete resections of pancreatic-head cancers has dropped significantly. As recently discovered, the fat surrounding the pancreatic head is infiltrated in most of the patients suffering from pancreatic head cancer. This presumably contributed to the low rates of complete resections. Therefore, these patients show signs of borderline resectability and may benefit from a chemotherapy prior to surgery. The aim of this study was to re-analyze the preoperative CT scans and to correlate those with the histopathological results. We found that the existence of cancerous infiltration of the fat surrounding the pancreas can be predicted by preoperative CT scan and that this in turn can discriminate between patients receiving complete or incomplete resections. Hence, a new standardized radiographic protocol should be implemented and preoperative chemotherapy may be warranted for at risk patients.

**Abstract:**

**Summary:** The rates of microscopic incomplete resections (R1/R0CRM+) in patients receiving standard pancreaticoduodenectomy for PDAC remain very high. One reason may be the reported high rates of mesopancreatic fat infiltration. In this large cohort study, we used available histopathological specimens of the retropancreatic fat and correlated high resolution CT-scans with the microscopic tumor infiltration of this area. We found that preoperative MDCT scans are suitable to detect cancerous infiltration of this mesopancreatic tissue and this, in turn, was a significant indicator for both incomplete surgical resection (R1/R0CRM+) and worse overall survival. These findings indicate that a neoadjuvant treatment in PDAC patients with CT-morphologically positive infiltration of the mesopancreas may result in better local control and thus improved resection rates. Mesopancreatic fat stranding should thus be considered in the decision for neoadjuvant therapy. **Background:** Due to the persistently high rates of R1 resections, neoadjuvant treatment and mesopancreatic excision (MPE) for ductal adenocarcinoma of the pancreatic head (hPDAC) have recently become a topic of interest. While radiographic cut-off for borderline resectability has been described, the necessary extent of surgery has not been established. It has not yet been elucidated whether pre-operative multi-detector computed tomography (MDCT) staging reliably predicts local mesopancreatic (MP) fat infiltration and tumor extension. **Methods:** Two hundred and forty two hPDAC patients that underwent MPE were analyzed. Radiographic re-evaluation was performed on (1) mesopancreatic fat stranding (MPS) and stranding to peripancreatic vessels, as well as (2) tumor diameter and anatomy, including contact to peripancreatic vessels (SMA, GDA, CHA, PV, SMV). Routinely resected mesopancreatic and perivascular (SMA and PV/SMV) tissue was histopathologically re-analyzed and histopathology correlated with radiographic findings. A logistic regression of survival was performed. **Results:** MDCT-predicted tumor diameter correlated with pathological T-stage, whereas presumed tumor contact and fat stranding to SMA and PV/SMV predicted and correlated with histological cancerous infiltration. Importantly, mesopancreatic fat stranding predicted MP cancerous infiltration. Positive MP infiltration was evident in over 78%. MPS and higher CT-predicted tumor diameter correlated with higher R1 resection rates. Patients with positive MP stranding had a significantly worse overall survival (*p* = 0.023). **Conclusions:** A detailed preoperative radiographic assessment can predict mesopancreatic infiltration and tumor morphology and should influence the decision for primary surgery, as well as the extent of surgery. To increase the rate of R0CRM− resections, MPS should be considered in the decision for neoadjuvant therapy.

## 1. Introduction

Ductal adenocarcinoma of the pancreatic head (hPDAC) is associated with a dismal prognosis, an overall 5-year survival rate of less than 5% and is estimated to become the second leading cause of cancer-related death by 2030 [1]. The only curative therapy that remains is surgical resection with an adjuvant treatment regime, advisably starting within 6 weeks after the operation [2,3]. Poor survival outcome in pancreatic cancer patients is partially explained by an advanced stage at initial diagnosis, and its consequent inaccessibility by surgery [4].

A standardized histopathological examination technique including the evaluation of the circumferential resection margin (CRM) was implemented in 2004, according to the recommendations of the Royal College of Pathologists [5,6]. Studies showed a significant influence of this technique on the margin-negative resection rate (R0). The medial pancreatic surface (groove of the portal vein/superior mesenteric vein and superior mesenteric artery) and the dorsal pancreatic resection margin (from inferior caval vein to abdominal aorta) remain the main sites for residual tumor. Positive resection margin rates in these locations are between 44–64% and 46–69%, respectively [6,7,8,9,10]. 

The degree of radicality during pancreatoduodenectomy for hPDAC has been a matter of some debate. However, since the introduction of the pathologic CRM staging exposed a lack of adequate margin negativity [6,9], it is obvious that surgical capacities have not been exhausted. Fortner et al. already described in the late 1970s an extended en-bloc regional resection, but failed to demonstrate a significant survival impact [11]. This may have been due to the scarcity of adjuvant chemotherapeutic regimes in that era. Only recently, the Japanese pancreatic society [12,13] was the first to introduce an extended standardized resection during pancreatoduodenectomy for hPDAC. These extended resections are surgically described as “mesopancreatic resections” or “excisions”. However, the “mesopancreas” as surgical-anatomical region is not yet conclusively defined and large anatomic studies to fully elucidate its impact are sadly lacking [14]. Nonetheless, this peripancreatic/mesopancreatic adipose region in immediate vicinity of the pancreatic tissue harbors an extensive amount of lymphatic tissue and perineural vessels. It was already postulated by Gockel et al. in 2007 to play a major role for R1 resections [15]. Unfortunately, studies further investigating this relationship [12,13,15,16] have been neglected in the past and extended resections are only performed in a few institutes in the Western world. 

Recently, we demonstrated the benefit of complete mesopancreatic excision (MPE) during structured pancreatoduodenectomy for hPDAC [16]. In over 78% of patients, the mesopancreatic fat was infiltrated. This infiltration was independent from the pathological T-stage, suggesting that a removal of the mesopancreatic fat is justified and feasible to secure local tumor control [16]. However, mesopancreatic fat infiltration was more abundant in R1 resected patients and those had a less favorable survival outcome [16], suggesting a possible benefit of a neoadjuvant treatment in this subgroup.

Multi-detector computed tomography (MDCT) of the abdomen remains the gold standard in preoperative diagnostics and staging for periampullary carcinomas. Previous studies have demonstrated the predictive value of portal vein (PV) infiltration in preoperative CT scans and predicted the need for PV resection upon surgery [17,18]. However, there is also data suggesting that preoperative staging tools often underestimate the local extent of tumors [19]. Considering the rates of margin negative resections for hPDAC and the implications of R1 or CRM+ resections on patient survival [16], as well as the advent of improved neoadjuvant treatments [20,21], better preoperative strategies to select patients most likely to be amenable to R0 resection are urgently needed. 

In MDCT, early and sparse tumor invasion of fatty tissue may be visible as “stranding”, an increased attenuation resulting from edema reminiscent of an inflammatory reaction. Based on the improved preoperative radiologic assessment, patients with even limited mesopancreatic fat infiltration and thus likely to receive R1 or CRM+ resections may be identified for neoadjuvant treatment followed by surgery, while others lacking those signs may benefit from a radical resection.

The aim of this study was to assess morphologic parameters in preoperative MDCT scans of hPDAC patients that predict mesopancreatic and vascular involvement and can therefore be used to better select patients that may benefit from a neoadjuvant chemotherapeutic approach.

## 2. Materials and Methods

### 2.1. Patient Selection and Demographic Data

Patients who had undergone partial pancreaticoduodenectomy with curative intent at the University Hospital of Duesseldorf between September 2003 and December 2020 were included for further evaluation, irrespective of tumor stage and microscopic resection margin status. In total, 343 patients suffering from PDAC were treated during the study period. Of these, 29 patients underwent oncologic distal pancreatectomy and were excluded from the study. In 72 patients, no preoperative MDCT scans were available for re-evaluation and thus, these patients were also excluded from the study. The remaining 242 patients met our inclusion criteria (108 females) (Appendix A). Clinicopathological and radiographic characteristics of the studied 242 patients are summarized in Table 1. The median age of all patients at the time of surgery was 70 years (range 41–95 years). Of the 242 patients, 193 (79.75%) patients presented without metastases (M0) and thus, received surgery with curative intent. In 49 (20.25%) patients, either a synchronous hepatic metastasis (*n* = 21, M1_(hep)_) or distant lymphatic para-aortic lymph node metastases (*n* = 28, M1_(PALN)_) were detected intraoperatively. No patient received neoadjuvant therapy, while 14 patients demonstrated vascular involvement currently classified as borderline resectable. TNM staging and grading were obtained from the original pathological reports. If necessary, the staging was updated to the 8th edition of the UICC TNM classification of Malignant Tumors [22] by experienced pancreatic pathologists (LH, IE). Clinical data regarding age at the time of surgery, gender, and overall survival were also reviewed. 

### 2.2. Radiographic Imaging

Patients were included if the scans of preoperative multiphasic multi-detector CT (MDCT) were available for re-evaluation (Appendix A). These examinations were retrospectively analyzed by three experienced hepatopancreaticobiliary radiologists (GF, GA, FZ) blinded for resection status and postoperative staging. To further minimize observer bias, scans from patients who did not meet the above-mentioned inclusion criteria and received MDCT for other reasons were re-analyzed as well and not included in the study. 

Each scan was re-evaluated and the following parameters were recorded: (1) Tumor diameter and distance to posterior and medial anatomic margins, and (2) mesopancreatic fat stranding. Furthermore, the presumed contact of the tumor to the superior mesenteric artery (SMA), common hepatic artery (CHA), gastroduodenal artery (GDA) portal, and superior mesenteric vein (PV/SMV) was analyzed and further sub-categorized by the circumferential degree of invasion (</>180°). For mesopancreatic stranding (MPS), the following grading system was applied: MPS 0: No fat infiltration, MPS 1: Infiltration of mesopancreatic fat < 2 mm, MPS 2: Infiltration > 2 mm without immediate major vessel contact, and MPS 3: >2 mm invasion of the mesopancreatic fat with immediate major vessel contact (Figure 1A–D and Figure 2A–D). 

### 2.3. Histopathological Analysis

Histopathological slides were re-evaluated by two experienced pancreatico-hepatobiliary pathologists (IE, LH) [16] (Figure 2A,C). Mesopancreatic fat invasion of the dorsal resection margin and the resection margin status were re-evaluated for each patient. If described in the histopathological report, cancerous infiltration of tissue surrounding the SMA and PV/SMV was obtained and compiled into a database. 

Standardized macroscopic and microscopic evaluation and reporting of pancreatic resection specimen, including CRM evaluation, were implemented at the University Hospital of Duesseldorf by September 2015. Between 2003 and September 2015, the resected specimens were examined without a standardized examination technique. Histopathological slides originating before 2015 were re-visited by a pathologist experienced in the hepatopancreaticobiliary system and if sufficient slides were available, a CRM status with evaluation of the mesopancreatic fat was evaluated. This included the evaluation not only of the dorsal, but also medial and ventral CRM. In addition, the “1-mm rule” was implemented for all patients according to the German oncology guidelines: A minimum margin clearance of 1 mm defines R0CRM−, whereas margin clearances between 0–1 mm are judged as R0CRM+ [23]. 

### 2.4. Surgical Therapy

All the resections were performed by trained pancreatic surgeons in our department. As recently described, a simultaneous mesopancreatic excision (MPE) followed by a para-aortic lymphadenectomy up to the right border of the SMA and circumferentially around the PV/SMV are obligatory components during pancreatoduodenectomy in our institution, see (11) for details. In summary, the aim of the procedure is a complete dissection of perineural and lymphatic tissue and structures dorsal to and surrounding the pancreatic head/uncinate process (Figure 3).

### 2.5. Statistical Analysis 

The Mann-Whitney U test and Pearson test were used to examine numerical data and to correlate between variables. For categorical data, the Chi-squared test and Fisher’s exact test were applied. Logistic regression analysis was applied for predication analysis, significant results are stated using hazard ratios and corresponding confidence intervals. Analyses were performed using SPSS^®^ statistics for Windows (version 26.0; SPSS, Inc., Chicago, IL, USA). A value of *p* < 0.05 was considered to indicate a statistically significant difference.

The study was carried out in accordance with the Good Clinical Practice, the Declaration of Helsinki, and an Institutional Review Board (IRB) approval of the Medical Faculty, Heinrich Heine University Duesseldorf (IRB no. 2019-473_1) was retrieved.

## 3. Results

### 3.1. Histopathological Results

The histopathological evaluation is summarized in Table 1. The CRM status and the fat tissue of the dorsal resection margin were evaluated in 197 patients (82.4%). Cancerous infiltration of the mesopancreatic fat was evident in 128 (65.0%) of these patients. True R0CRM− resections were performed in 86 of 197 (35.5%) patients, R0CRM+ and R1 resections were achieved in 48 (19.8%) and 63 (26.0%) patients, respectively. All correlation and prediction analysis were performed with the 197 patients with CRM resection status (R0CRM− vs. R0CRM+/R1 or R0CRM−/R0CRM+ vs. R1) (Table 1).

In 97/242 (40.1%) of these patients, simultaneous portal vein resection was performed during PD. In 64 (26.4%) of these patients without portal vein resection, the tissue surrounding the PV was harvested during surgery and was separately investigated in the pathological reports (Appendix A). In 39 of these 161 patients (24.2%) with histopathological evaluation of the PV, tumor infiltration of the portal vein was evident. 

In 77/242 (31.8%) of these patients, the resected tissue surrounding the SMA was available for evaluation. In 14 (18.2%) of these patients, tumor infiltration in the peri-arterial tissue was detected. None of the patients in this cohort received arterial resection or reconstruction. 

### 3.2. Radiographic Results

All the radiographic variables are summarized in Table 1. A presumed malignant mass was detected in all 242 patients (Figure 1A–D). The median diameter was 25 mm (range: 7–60 mm). The distance to dorsal plane (IVC/AA) was 5 mm (median, range: 0–25 mm). Tumor contact to the CHA, GDA, SMA, and PV/SMV was evaluated. Tumor contact to the PV/SMV was detected in 89 patients (36.8%). Tumor contact was detected to the GDA in 60 patients (24.8%), to the SMA in 27 patients (11.2%), and to the CHA in seven patients (2.9%) (Table 1).

Mesopancreatic stranding (MPS) dorsal to the head of the pancreas was sub-grouped as described in Materials and Methods (Figure 1A–D). In only 60 patients, MPS was not visible (24.8%), whereas MPS 1, 2, and 3 were found in 69 (28.5%), 33 (13.6%), and 80 (33.1%) patients, respectively. Out of the 80 MPS, isolated peri-vascular fat stranding around the PV/SMV was detected in eight patients (10.0%), whereas isolated fat stranding around the SMA was detected in 18 patients (22.5%). In 39 patients (48.8%), both PV/SMV and SMA showed a simultaneous fat stranding. Fat stranding around the GDA and CHA was visible in 10 (12.5%) and in five patients (6.25%), respectively (Table 1 and Figure 4B).

Peri-vascular fat stranding around the PV/SMV was detected in 48 patients (19.8%), whereas fat stranding was detected around the SMA in 58 patients (24.0%), followed by the GDA in 10 (4.1%), and CHA in five patients (2.1%) (Table 1).

### 3.3. Correlation Analysis of Radiographic and Histopathological Results

The T-stage was available for the complete study cohort (*n* = 242). For 197 patients, mesopancreatic fat specimen and CRM status were available for correlation with radiographic variables and resection status. Analysis of PV (*n* = 161) and SMA (*n* = 77) infiltration was performed in patients where separate histopathological reporting was available (Table 2).

Tumor diameter and tumor distance to the dorsal plane (ICV/AA) in MDCT significantly correlated with the pathological T-stage (*p* < 0.001 and 0.010) (Figure 5A,B and Table 2). MDCT detected that fat stranding at the dorsal plane correlated significantly with pathologic mesopancreatic tumor infiltration at the dorsal resection margin (*p* = 0.001) (Table 2). Both MDCT detected tumor contact and peri-vascular fat stranding (MPS 3) to the SMA and PV/SMV correlated significantly with the pathologic infiltration of these structures (*p <* 0.001 and *p =* 0.011 for tumor contact around the SMA and PV, respectively; *p =* 0.006 and *p =* 0.037 for MP fat stranding around the SMA and PV, respectively) (Table 2). 

In the 197 patients with histological resection status including CRM, the correlation of complete resection (R0CRM−) and incomplete resection (R1 or R0CRM+) with radiographic variables was evaluated (Table 3). Out of the MDCT variables, tumor diameter and positive MPS significantly correlated with the R1/R0CRM+ resection status (Figure 6 and Table 3).

Histologically evident mesopancreatic fat infiltration correlated with a significantly higher rate of R1/R0 CRM+ resections when compared to patients without mesopancreatic fat infiltration (Table 4). Radiographic MPS was detected in 144 of 197 patients (73.1%), of these 144 patients, R0(CRM−) resections were achieved in only 38.9% (56 patients). Interestingly, in the 53 MPS negative patients, R0(CRM−) resections were significantly more prevalent with 56.6% (30 patients) (*p =* 0.010) (Table 4).

### 3.4. Survival Analysis

Of all the 193 M0 resected patients, complete datasets including CRM status and preoperative MDCT were available in 153 patients, and those were included in the gross survival analysis (Table 5). Sixteen patients deceased during the first 30 postoperative days (mortality rate 6.6%). The median OS of all the 153 M0 resected patients was 1.603 years (95% CI: 1.170–2.036 years). 

In the univariate analysis of the whole M0 cohort (*n* = 153), the following clinicopathological parameters were associated with prognostic impact: Median age, resection margin, multidrug chemotherapeutic regime, and mesopancreatic fat stranding (Table 5 and Figure 7A). In multivariate analysis of the whole M0 cohort, only the negative resection margin (R0(CRM−)) remained as an independent prognostic factor (Table 5).

A further survival analysis was performed for the 69 R0(CRM−) resected M0 patients. Of these, 24 patients had no evidence of MPS in their preoperative MDCT. Of the 45 patients with MPS, 19 patients were graded as MPS1, whereas 6 and 20 patients were graded as MPS2 and MPS3, respectively. 

The Kaplan-Meier survival analysis of M0 patients with (*n* = 45) and without (*n* = 24) MPS demonstrated a significantly longer overall survival in MPS negative patients (median OS 2.89 years, 95% CI 1.88–3.89) compared with MPS positive patients (median OS 1.29 years, 95% CI 0.59–1.98) (*p = 0.025*) (Figure 7B). 

Survival analysis in the 84 R0(CRM+)/R1 resected patients revealed no prognostic significance when stratified according to the MPS status (MPS 0 vs. 1–3 (Figure 7C). The median OS of 1.22 years (95% CI 0.0–2.65) in group MPS 0 (*n* = 21) was similar compared with MPS 1–3 patients (*n* = 63) (median OS 1.28 years, 95% CI 0.87–1.69) (*p =* 0.436).

## 4. Discussion

Preoperative MDCT can reliably predict tumor extension and fat infiltration of the mesopancreas and these variables correlate well with surgical resection status and overall survival outcome in patients with primary resectable hPDACs. 

The aim of this study was to test the reliability of preoperative MDCT to predict histopathological infiltration of the mesopanreatic fat and to assess morphologic parameters that predict mesopancreatic and vascular involvement. Mesopancreatic fat infiltration has recently gained attention in complete resection of PDAC [16], with survival outcome and the likelihood of complete R0(CRM-) resection. A more reliable preoperative assessment will allow an individualized treatment approach and possibly improve outcomes. 

Despite numerous publications on MDCT and PDAC, it has so far not been reported if MDCT-estimated tumor size correlates with the redefined size-based T-stage of the 8th TNM classification [18,19,24,25,26,27]. In our study, the MDCT-presumed tumor size correlated with, and reliably predicted, histological pT-stage. Furthermore, we observed a significant relationship between the larger MDCT predicted tumor size and closer relationship of the tumor to the dorsal ICV/AA plane. 

Mesopancreatic tissue—or the retropancreatic lamina—was first defined in 2008 by Gockel et al. [15]. It is postulated to play a major role in R1 resections of hPDAC [16,28]. Although in the study cohort, a complete mesopancreatic excision (MPE) was performed, tumor free resection margins (R0CRM-) were achieved in ~50% of the patients. Compared with resection rates previously reported [6], these R0CRM- rates are high, but compared with resection rates in other malignancies of the gastrointestinal tract [29], these rates need to be improved. 

In our study, MDCT-presumed radiographic fat stranding predicted mesopancreatic infiltration. Thus, MDCT-presumed fat stranding itself cannot be regarded as a simple desmoplastic reaction or edema and should be used for decision making in multimodal therapeutic approaches [30,31].

Re-resection of intraoperatively detected positive resection sites has been demonstrated to have only a marginal influence on survival, which further highlights the importance of primary margin negative resections (R0CRM-) in PDAC surgery [32,33,34]. The persistently high rates of margin positive resections suggest that the necessary extent of surgery has not been achieved over the past decade [6,9]. These high rates of positive resection margins may also reflect an underestimation of the close anatomical relationship of the tumor and the dorsal fat plane. This is also highlighted by the high incidence of mesopancreatic fat infiltration [16]. This microscopic infiltration of the soft mesopancreatic fat renders a R0CRM- resection unlikely. In our cohort of radically resected patients, mesopancreatic fat infiltration correlated with a significantly higher rate of positive resection status (R1 and R1 + R0(CRM+)). 

As an individualized therapy becomes the norm, neoadjuvant therapy for borderline resectable PDACs with involvement of the peripancreatic vessels is becoming more prevalent [35,36,37]. While the vascular involvement is likely an indicator of an unfavorable tumor topography rather than an adverse tumor biology [38], reports on resection margin rates and survival with respect to vascular involvement are hampered by surgical heterogeneity [35,36,39,40,41,42,43]. 

We previously reported on the influence of margin negative resections (R0(CRM−)) on the prognosis in hPDAC patients and demonstrated the importance of complete mesopancreatic excision [16]. In this study, we performed a similar survival analysis in patients with both CRM status and preoperative MDCT. In univariate analysis, positive MPS (MPS 1–3) and positive resection margins (R1/R0(CRM−)) were prognostic factors for OS. In multivariate analysis, again only R0(CRM−) resection was left as an independent prognostic factor for OS, highlighting the importance of primary margin negativity during surgery for hPDACs. 

However, non-metastasized R0(CRM−) patients with radiographic MPS had a significantly shorter median overall survival, while the amount of MPS did not seem to matter (MPS 1–3). Therefore, we conclude that the radiographic assessment of MPS may allow the selection of patients with presumably more aggressive tumor biology, even if resected extensively. Margin negativity remains the most important factor for prolonged survival, which is corroborated by our observation that MPS did not stratify the survival of R0(CRM+)/R1 resected patients. The decision for multimodal therapeutic regimes (neoadjuvant vs. upfront surgery) is to date solely based on vascular affection. In order to significantly increase surgical margin clearance, MPS as an independent factor, could play a crucial role for treatment stratification.

Based on the evidence presented, we suggest that primary surgical resection of PDAC should be limited if the mesopancreatic dissection plane is radiographically presumed to be infiltrated [12,16], similar to patients with peripancreatic vascular involvement. By including tumor diameter and MPS in the standardized preoperative MDCT evaluation of resectability, a higher margin negative resection rate is likely to be achieved in primary resected PDAC. Patients who have been identified as borderline resectable due to MPS should also benefit from a preoperative chemotherapeutic approach. In this study, radiographic evaluated MPS and histopathologically detected mesopancreatic fat infiltration correlated significantly, as did mesopancreatic fat infiltration and R1/R0CRM+ resection. This emphasizes the role of a detailed preoperative work-up to identify patients which may be more suitable for a neoadjuvant chemotherapeutic approach. Prospective multi-centric trials are therefore clearly warranted to further elucidate the benefit of neoadjuvant treatment of patients with MPS+ PDAC.

## 5. Conclusions

A structured preoperative MDCT assessment can adequately predict infiltration of the mesopancreatic fat and peripancreatic vessels, tumor size, and tumor location. Any involvement of the mesopancreatic fat (MPS 1–3) was a predictor for worse OS even in R0(CRM-) patients and should be considered an independent marker for inclusion in multimodal treatment regimens. Patients with a higher T-stage and/or positive MPS may be amenable to neoadjuvant treatment regimens, in order to achieve higher rates of surgical margin clearance. Prospective trials are warranted to further elucidate the benefit of multimodal treatment regimens in patients with radiographic MPS. 

## Figures and Tables

**Figure 1 cancers-13-04361-f001:**
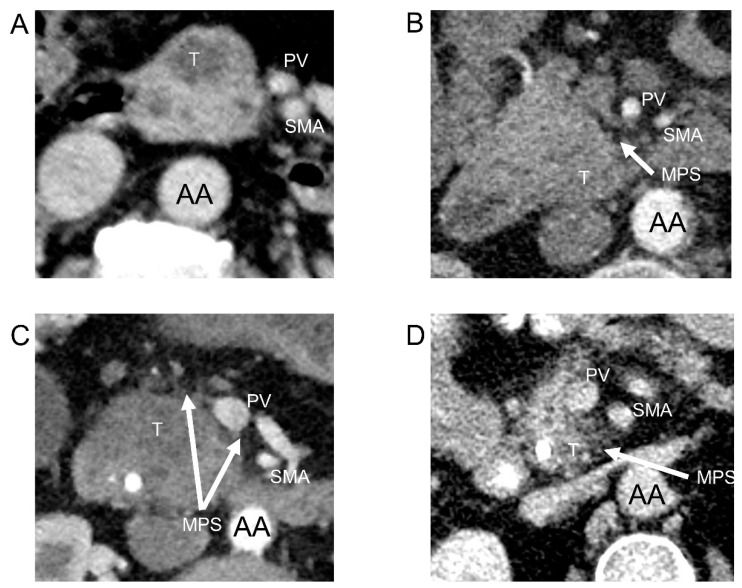
(**A**) MDCT slide without mesopancreatic fat stranding; (**B**) MDCT slide with MPS 1; (**C**) MDCT slide with MPS 2; (**D**) MDCT slide with MPS 3 (AA: Abdominal aorta; MPS: Mesopancreatic fat stranding; PV: Portal vein; SMA: Superior mesenteric artery; T: Tumor).

**Figure 2 cancers-13-04361-f002:**
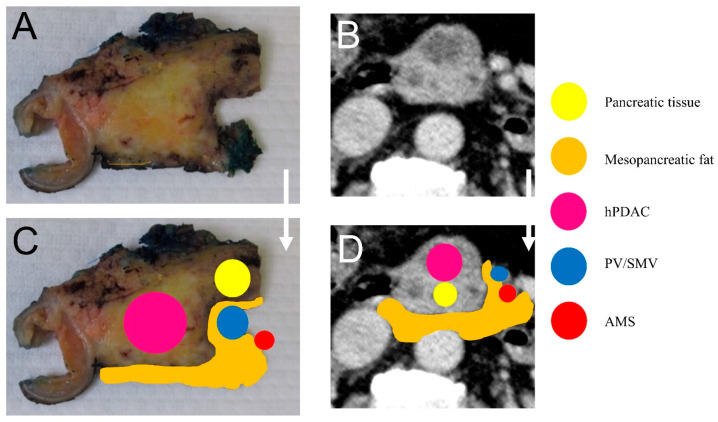
(**A**) Pathological specimen of the pancreatic head with infiltration of the peripancreatic fatty tissue. The specimen was inked using a pre-defined color code (posterior surface: Black; anterior surface: Blue; medial surface: Green). Grossing was done according to the axial slicing technique (here: pT3 pN2 (5/47) L1 V0 Pn1); (**B**) MDCT slide without MPS; (**C**) edited pathological specimen visualizing the hPDAC as well as the mesopancreatic fat; (**D**) edited MDCT slide without MPS visualizing the hPDAC as well as the mesopancreatic fat (hPDAC: Ductal adenocarcinoma of the pancreatic head; MPS: Mesopancreatic fat stranding; PV: Portal vein; SMA: Superior mesenteric artery; SMV: Superior mesenteric vein).

**Figure 3 cancers-13-04361-f003:**
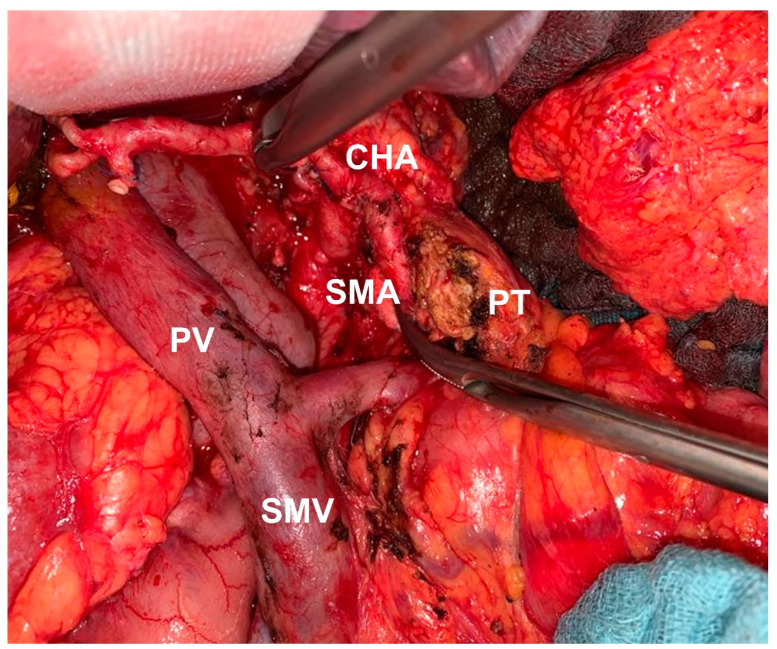
Operative situs after mesopancreatic excision during PD (CHA: Common hepatic artery; PV: Portal vein; PT: Pancreatic tail; SMA: Superior mesenteric artery; SMV: Superior mesenteric vein).

**Figure 4 cancers-13-04361-f004:**
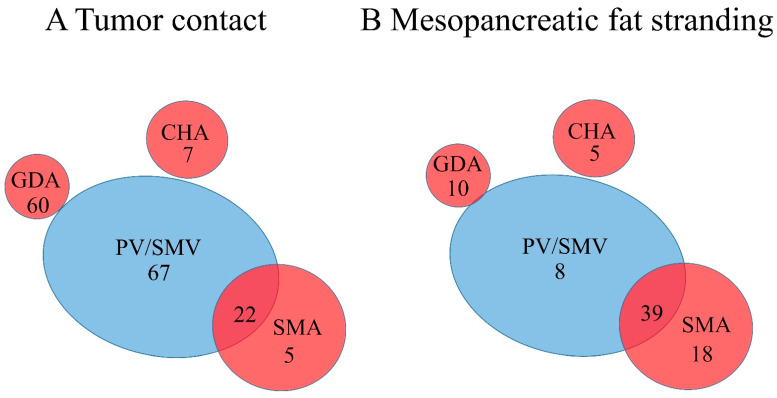
(**A**) Illustration visualizing separate and synchronous histological tumor contact to peripancreatic vessels. (**B**) Illustration visualizing separate and synchronous mesopancreatic fat stranding (MPS 3) to peripancreatic vessels (CHA: Common hepatic artery; GDA: Gastroduodenal artery; MPS: Mesopancreatic fat stranding; PV: Portal vein; SMA: Superior mesenteric artery; SMV: Superior mesenteric vein).

**Figure 5 cancers-13-04361-f005:**
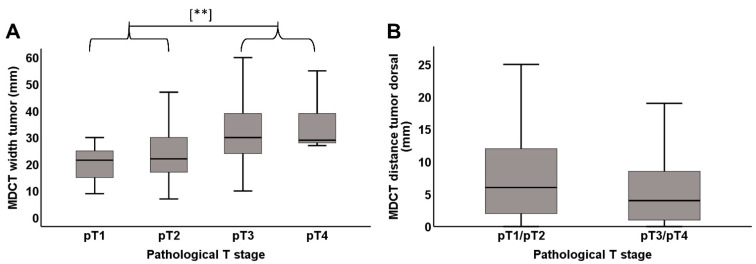
(**A**) Box plot of radiographically assumed tumor width and pathological T-stage. Pearson test was used to test for statistical difference between pT1+2 vs. pT3+4 (*p =* 0.001) ** indicates a *p*-value ≤ 0.01. (**B**) Box plot of radiographically assumed tumor distance to dorsal margin (ICV/AA) in relation to the pathological T-stage. Pearson/spearman test was used to test for statistical significance (*p =* 0.011).

**Figure 6 cancers-13-04361-f006:**
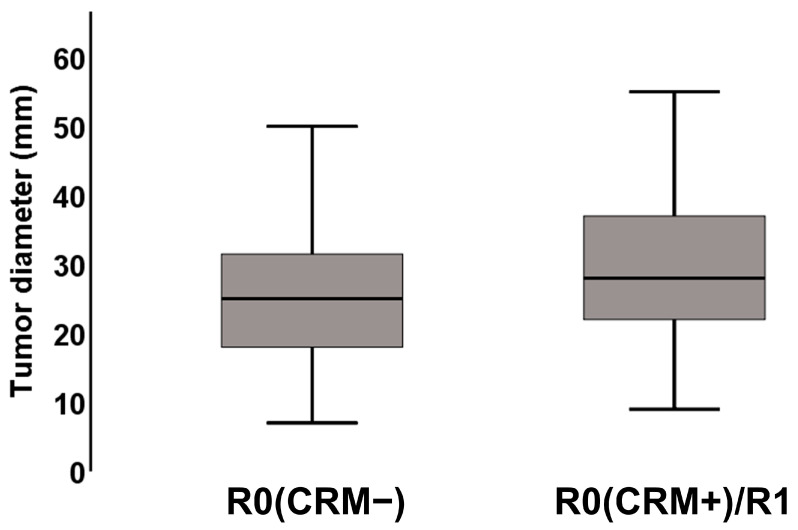
Box plot of MDCT-presumed tumor diameter and resection status. Pearson/spearman test was used to test for statistical significance (*p* = 0.033).

**Figure 7 cancers-13-04361-f007:**
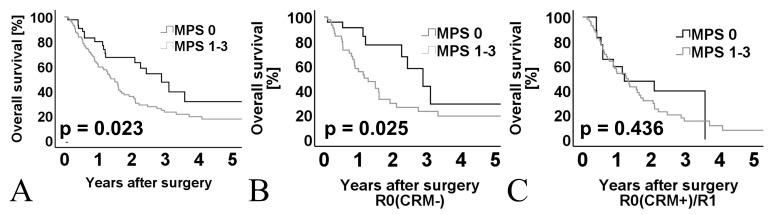
(**A**) Kaplan-Meier curve for OS of patients with and without MPS of the entire cohort, *n* = 153. (**B**) Kaplan-Meier curve for OS of patients with and without MPS of R0(CRM-) resected patients, *n* = 69. (**C**) Kaplan-Meier curve for OS of patients with and without MPS of R0(CRM+)/R1 resected patients, *n* = 84. Log-rank test was used to test for significance.

**Table 1 cancers-13-04361-t001:** Demographic table of all 242 included patients. Staging is revised to the 8th edition of the UICC TNM classification of malignant tumors.

Age in Years Median (Range)	70 (41–95)		Tumor Width		
			Median (range)	25 mm (7–60 mm)	
			Distance from dorsal margin		
	*n*	%	Median (range)	5 mm (0–25 mm)	
Sex					
Male	134	55.4		*n*	%
Female	108	44.6	Tumor contact in MDCT		
T-stage			SMA	27	11.2
T1	15	6.2	Contact > 180°	20	8.3
T2	137	56.6			
T3	85	35.1	CHA	7	2.9
T4	5	2.1	Contact > 180°	6	2.5
N-stage					
N0	39	16.1	GDA	60	24.8
N1/2	203	83.9	Contact > 180°	32	13.2
M-stage					
M0	193	79.8	PV/SMV	89	36.8
M1	49	20.2	Contact > 180°	28	11.6
Grading					
G1/G2	136	56.2	MPS in MDCT		
G3	102	42.1	positive MPS	182	75.2
missing	4	1.7			
Pn			stranding to SMA	58	24.0
Pn0	50	20.7			
Pn1	183	75.6	stranding to CHA	5	2.1
missing	9	3.7			
L			stranding to GDA	10	4.1
L0	116	47.9			
L1	117	48.3	stranding to PV/SMV	48	19.8
missing	9	3.7			
V					
V0	170	70.2			
V1	63	26.0			
missing	9	3.7			
R-status (CRM)					
R0CRM−	86	35.5			
R0CRM+/R1	111	45.8			
missing	45	18.6			

CHA: Common hepatic artery; CRM: Circumferential resection margin; GDA: Gastroduodenal artery; ICV: Inferior caval vein; L: Lymphatic invasion; LN: Lymph nodes; MPS: Mesopancreatic fat stranding; Pn: Perineural invasion; PV/SMV: Portal/superior mesenteric vein; SMA: Superior mesenteric artery; V: Venous invasion.

**Table 2 cancers-13-04361-t002:** Correlation analysis of radiographic and histopathological variables. Statistical difference was calculated by Pearson analysis. Logistic regression analysis was performed for prediction assessment.

Histopathology		MDCT Scan		*p-*Value	Sensitivity/ Specificity	HR	95%CI
		Tumor Width (mm)					
T-Stage	*n*	Median	Range				
pT1	15	21	9–30	<0.001	40%/75%	1.690	1.2–2.3
pT2	137	23	7–50		30%/83%		
pT3	85	30	10–60		57%/71%		
pT4	5	29	27–55		60%/89%		
**Modified Contingency Tables** **Tumor Morphology**							
**Histopathology**		**MDCT Scan**					
**PV/SMV Infiltration**		**PV/SMV Tumor Contact**					
	***n***		***n***	<0.001	77%/74%	9.375	4.1–21.9
Yes	39	Yes	30 of 39				
No	122	No	90 of 122				
**SMA Infiltration**		**SMA Tumor Contact**					
	***n***		***n***	0.010	43%/89%	5.893	1.6–22.0
Yes	14	Yes	6 of 14				
No	63	No	56 of 63				
**Modified Contingency Tables** **Mesopancreatic Fat**							
**Histopathology**		**MDCT Scan**					
**MP Fat Infiltration**		**MPS**					
	***n***		***n***	0.001	80%/41%	2.709	1.4–5.3
Yes	128	Yes	103 of 128				
No	69	No	28 of 69				
**PV/SMV Infiltration**		**MPS to PV/SMV**					
	***n***		***n***	0.037	12%/71%	NS	NS
Yes	39	Yes	5 of 39				
No	122	No	86 of 122				
**SMA Infiltration**		**MPS to SMA**					
	***n***		***n***	0.006	71%/70%	5.789	1.6–20.8
Yes	14	Yes	10 of 14				
No	63	No	44 of 63				

CI: Confidence interval; HR: Hazard ratio; MP: Mesopancreatic; MPS: Mesopancreatic fat stranding; PV/SMV: Portal/superior mesenteric vein; SMA: Superior mesenteric artery.

**Table 3 cancers-13-04361-t003:** Correlation analysis of histopathological mesopancreatic fat infiltration and resection status. Statistical difference was calculated by Fisher’s exact test.

	Resection Status R0CRM− vs. R1/R0CRM+
Radiographic Variable	*p-*Value
</≥ 2 cm tumor diameter	0.048
</≥ median tumor distance dorsal plane (AA/ICV)	0.339
contact SMA yes/no	1.000
contact SMA > 180° yes/no	0.302
contact PV/SMV yes/no	0.149
contact PV/SMV > 180° yes/no	1.000
MPS yes/no	0.010
stranding to SMA yes/no	0.731
stranding to PV/SMV yes/no	0.057

CI: Confidence interval; HR: Hazard ratio; MPS: Mesopancreatic fat stranding; PV/SMV: Portal/superior mesenteric vein; SMA: Superior mesenteric artery.

**Table 4 cancers-13-04361-t004:** Correlation analysis of radiographic variables and resection status. Statistical difference was calculated by Pearson analysis. Logistic regression analysis was performed for prediction assessment.

Mesopancreatic Fat Infiltration and Resection Status						*p*-Value Fisher Exact Test
Histopathological MP Fat Infiltration		Resection Status R0CRM− vs. R1/R0CRM+				
	***n***					
yes	128	R1 or R0CRM+ rate		70.3%		<0.001
no	69	R1 or R0CRM+ rate		30.4%		
**Resection Status**						
		**R1** ***n***	**R0(CRM+)** ***n***		**R0(CRM-)** ***n***	0.010
MPS	yes	52	36		56	
	no	11	12		30	

CRM: Circumferential resection margin; MPS: Mesopancreatic stranding; *n*: Number.

**Table 5 cancers-13-04361-t005:** Univariate and multivariate survival analyses for overall survival of all M0 resected patients, n = 153. Analyses were performed by the log-rank test and cox logistic forward regression.

Univariate Analysis				
		*p*-Value		
Median age (< vs. > median)		0.003		
T-stage (T1/T2 vs. T3/T4)		0.223		
N-stage (N0/N1 vs. N2)		0.455		
Grading (G1/G2 vs. G3)		0.109		
Pn (Pn0 vs. Pn1)		0.824		
L (L0 vs. L1)		0.643		
V (V0 vs. V1)		0.164		
R-status (R0(CRM−) vs. R1/R0(CRM)+)		0.002		
Gemcitabine mono vs. Multidrug CTx		0.049		
MPS (MPS 0 vs. MPS 1–3)		0.023		
**Multivariate Analysis**				
	***p-*Value**		**HR**	**95%CI**
R-status (R0(CRM-) vs. R1/R0(CRM)+)	0.047		1.592	1.006–2.519

CTx: Chemotherapy; CI: Confidence interval; HR: Hazard ratio; MPS: Mesopancreatic fat stranding; multidrug: Gemcitabine based or FOLFIRINOX; L: Lymphatic invasion; Pn: Perineural invasion; V: Venous invasion.

## Data Availability

Data available on request due to restrictions (privacy and ethical). The data presented in this study are available on request from the senior authors.

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
