# Peer review of "Pre-Operative MDCT Staging Predicts Mesopancreatic Fat Infiltration—A Novel Marker for Neoadjuvant Treatment?"

_cancers, 2021, doi:10.3390/cancers13174361_

Round 1

Reviewer 1 Report

The authors responded sufficiently to the comments raised by the reviewers and the manuscript has improved substantially. I have no more comments.

This manuscript is a resubmission of an earlier submission. The following is a list of the peer review reports and author responses from that submission.

Round 1

Reviewer 1 Report

Safi and coworkers present results from a large retrospective patient cohort on the predictive value of a novel evaluation system for preoperative CT-scans for adenocarcinoma of the head of the pancreas. 

They report evidence for the correlation between radiological signs of infiltration of the mesopancreas, tumour size and location. The radiological involvement of the mesopancreas proved to be predictive for impaired survival in patients where an R0-resection with a margin of more than 1mm could be achieved.

The study applied  a novel approach to a sizeable cohort, making it a very valuable contribution to prognostication efforts. The visual material to illustrate radiological sign and surgical procedure is excellent and the procedural descriptions are very clear. Furthermore, the complexity that is to be expected in an unselected cohort is well adressed by selection of relevant subgroups. However, the following points should be addressed:

General comments:

  1. Introduction, line 77: I would commend a brief mention of previous work like the first description by Gockel and the recommendations in current clinical guidelines on the value of MPE.
  2. Methods: MD-CT has has undergone a rapid technical evolution since the start of the study. The authors should briefly address if and how increasing detection array size and other technical progress has further facilitated the determination of MPS categories or infiltration. 
  3. Results: In the survival analysis, R-status remained the only independent predictor. So, MPS-status was seemingly eliminated in the multivariable regression. But the KM-curves in Fig5 show a significant impact of the MPS-status. Please explain.
  4. For the appreciation of the impact of MOS-categories it would be beneficial to present KM-curves  both for the entire group of resected patients  as well as for the R0CRM- and the R1/R0CRM+ subgroups. The authors further chose to address the impact of MPS-status both by dichotomy for present/absent and for medium/strong. These groups overlap (MPS1 cases "changing sides"), making the interpretation difficult. Do the data support considering MPS0, MPS 1 and MPS2/3 as separate prognostic entities?  The argument in line 365 indicates otherwise and is partially contradictory.
    The rationale for presenting the medium to strong MPS-subgroups and its implication should be either made clearer or preferably be omitted.

Minor corrections

  1. Line 57: regimen (instead of regime)
  2. Figure SI: CRM was omitted in the legend
  3. Table 1:  Gender is a sociocultural concept, use Sex to refer to biological differences.
  4. Table 2: NS in SV/SM infiltration: please explain to the legend. It is not obvious why neither HR or CI could be computed although p and specificity/sensitivity were determined?
  5. Table 3: AA/IVC not included in the legend
  6. Fig5 A and B seem to be exchanged. The legend in B MPS1+2 should read MPS 2+3,  the risk-table below is correct
  7. Line 323: Convoluted sentence, please simplify. Does the sentence refer to the R-status of the current study or the cohort in the reference?
  8. Line 327: The sentence should state the sensitivity of MCT for detection of mesopancreatic infiltration

Reviewer 2 Report

According to the authors, the aim of the study was to evaluate the ability of preoperative MDCT to predict the hispathologically proven mesopanreatic fat infiltration by the tumor. Nevertheless, in the introduction they state that “The aim of this study was to assess morphologic parameters in preoperative MDCT scans of hPDAC patients that predict mesopancreatic and vascular involvement”.

From the materials and methods, it is apparent that many other variables such as lymph node and vessel involvement have been included. Apparently many tumors were borderline resectable but there is no information if these patients had neo-adjuvant chemotherapy. The time span between September 2003 and December 2020 is very long and many changes in the quality of imaging but also surgery and treatment improvements occured during this period. A standardized examination technique of the resected specimens and histopathology reporting was introduced after September 2015. The terms “mesopancreas”, “dorsal plane”, “dorsal resection margin” should be clearly defined for readers unfamiliar with pancreatic surgery. A clear definition of the radiographic variables should be also included. All these factors introduce some heterogeneity and might have affected the results.

The results are overloaded and do not serve the main aim of the study. Their presentation should be clear and succinct. In the footnote of Figure SI is written “MPI: mesopancreatic infiltration” but this is not shown in the figure. Similarly, in Tables 3 and 5 no Hazard ratios and Confidence intervals are shown as stated in the footnotes, implying uni- and multi-variate analyses. Hazard ratios along with 95% Confidence intervals should be also shown for univariate analysis in Table 6.

A plain study design with straightforward comparisons of preoperative MDCT findings regarding mesopancreatic fat stranding and histological findings of resected mesopancreatic tissue might serve better the aim of the study.